# Green Synthesis of CuO Nanoparticles from Macroalgae *Ulva lactuca* and *Gracilaria verrucosa*

**DOI:** 10.3390/nano14131157

**Published:** 2024-07-06

**Authors:** Marta Marmiroli, Marco Villani, Paolina Scarponi, Silvia Carlo, Luca Pagano, Valentina Sinisi, Laura Lazzarini, Milica Pavlicevic, Nelson Marmiroli

**Affiliations:** 1Department Chemistry, Life Sciences, and Environmental Sustainability, University of Parma, Parco Area delle Scienze, 43124 Parma, Italy; paolina.scarponi@unipr.it (P.S.); milica.pavlicevic@unipr.it (M.P.); 2Istituto dei Materiali per l’Elettronica ed il Magnetismo (CNR IMEM), Parco Area delle Scienze, 43124 Parma, Italy; marco.villani@imem.cnr.it (M.V.); valentina.sinisi@imem.cnr.it (V.S.); laura.lazzarini@imem.cnr.it (L.L.); 3Consorzio Interuniversitario Nazionale per le Scienze Ambientali (CINSA), University of Parma, Parco Area delle Scienze, 43124 Parma, Italy; luca.pagano@unipr.it (L.P.); nelson.marmiroli@unipr.it (N.M.)

**Keywords:** green macroalgae, red macroalgae, CuO nanoparticles, bactericidal, fungicidal

## Abstract

Macroalgae seaweeds such as *Ulva lactuca* and *Gracilaria verrucosa* cause problems on the northern coast of the Italian Adriatic Sea because their overabundance hinders the growth of cultivated clams, *Rudatapes philippinarum*. This study focused on the green synthesis of CuO nanoparticles from *U. lactuca* and *G. verrucosa*. The biosynthesized CuO NPs were successfully characterized using FTIR, XRD, HRTEM/EDX, and zeta potential. Nanoparticles from the two different algae species are essentially identical, with the same physical characteristics and almost the same antimicrobial activities. We have not investigated the cause of this identity, but it seems likely to arise from the reaction of Cu with the same algae metabolites in both species. The study demonstrates that it is possible to obtain useful products from these macroalgae through a green synthesis approach and that they should be considered as not just a cause of environmental and economic damage but also as a potential source of income.

## 1. Introduction

Nanotechnology can blend biology principles with physics and chemistry to generate nano-components having specific functions [1,2,3,4,5,6]. Nanoparticles (NPs) exhibit different sizes and shapes, but diameters typically range between 1 and 100 nm. Compared to raw bulk material, NPs show unique physico-chemical properties due to their high surface area to volume ratio. Reduced cohesive energy, and a higher degree of curvature, enable NPs to act as catalysts for surface-sensitive reactions [7].

Such remarkable characteristics give rise to novel opportunities in different fields, including therapeutics, drug discovery, optoelectronics, diagnostic biological probes, display instruments, catalysis, sensors, and detection of toxic metals or other environmental contaminants [8,9,10]. Most nanoparticles are prepared by inorganic synthesis, but it is also possible to synthesize them utilizing natural products such as plants and algae, and bacterial and fungal extracts [11].

NPs can be synthesized by two fundamental methods: top-down and bottom-up. Within the top-down approach, NPs are generated by reducing the size of the bulk material, employing several physical and chemical methods [12,13,14]. These microfabrication techniques include laser ablation, etching, sputtering, mechanical milling, and electro-explosion [15].

The bottom-up approach produces NPs by the assembly of atoms, molecules, and clusters. Hence, it is also referred to as “molecular nanotechnology” [15,16]. The nano-sized structures produced by the bottom-up approach are created by methods such as chemical reduction, plasma or flame spraying, sol-gel processes, molecular condensation, supercritical fluid synthesis, laser pyrolysis, use of templates, chemical vapor deposition and, most significantly, by alternative biologically-based green synthesis [17,18,19,20].

Compared to chemical methods, biological materials are in high demand for NP synthesis. A wide variety of bacteria, fungi, yeasts, marine and freshwater algae, and plants have been utilized for NP synthesis because they are eco-friendly and low cost: these processes are called “nanobiofactories” [21].

Macroalgae (seaweeds) are often used as a potential source of secondary metabolites, including phenolic compounds, pigments, and polysaccharides [22]. Biosynthesis based on the abilities of macroalgae as nanobiofactories targets algal secondary metabolites for use as reducing agents to stabilize NPs; most studies have been focused on the production from algae of metal (Ag, Au) and metal-oxide (CuO, ZnO) NPs: the eco-friendly biosynthesis of metal NPs using bioactive compounds from macroalgae reduces cost and production time and increases their biocompatibility, making them suitable for a wide variety of applications [23,24].

Cell wall components of brown macroalgae, such as polysaccharides (e.g., alginates and fucose-containing sulfated polysaccharides), are functional to green biosynthesis [22]. Red macroalgae extracellular matrices contain sulfated galactans, agars, and carrageenans. In comparison, marine green macroalgae extracellular matrices include different types of polysaccharides (e.g., semicrystalline cellulose, water-soluble ulvans, and two minor hemicelluloses) [25]. Seaweeds also contain pharmacologically active substances such as alkaloids, terpenoids, flavonoids, and phenols. Marine brown, red, and green macroalgae have significant differences in their physiological and intracellular biological contents. A comprehensive description of the constituents of brown, red, and green macroalgae is reported by Kloareg et al. [26].

These properties and their abundance as a raw material have attracted many researchers to consider their use in cleaner methods for NP synthesis [27]. There is a recent trend in nanotechnology to evaluate possible synergic effects between the nanomaterials being produced and natural biomolecules. Among these, natural antioxidants have attracted considerable attention since they can act against oxidative stress, which has been shown to be an important factor in the appearance and evolution of many human diseases, which include diabetes, cardiovascular diseases, cancer, and even aging [28]. Algae are photoautotrophic and produce much of the world’s oxygen. They can also bioaccumulate heavy metals. Antimicrobial activity is a desired property of new biological synthesized nanomaterials. Seaweeds can potentially reduce oxide materials to antibacterial metallic nanoparticles: the first metallic nanoparticles, with their unique antibacterial properties, were produced using *Sargassum weightii* [29].

Very little is known about the biological impact of green-synthesized NPs, and in particular about toxicity, uptake, and bioaccumulation mechanisms. Compared with chemically synthesized NPs, green-synthesized NPs are not widely used in industry. Antimicrobial resistance is one of the major global threats for human health, and it is considered by the World Health Organization as a priority issue [30,31]. Bacterial and fungal infections of humans are common, and the treatment is becoming increasingly challenging due to increasing resistance to standard antimicrobials [32]. Among pathogens identified as priorities by the World Health Organization (WHO) in its first Fungal Priority Pathogens List—WHO FPPL, published in 2022—*Candida albicans* and *Candida auris* are included in the priority group [33,34]. *C. albicans* is able to acquire resistance to the commonly used antifungal agents (e.g., fluconazole). The exponential development of multidrug resistance, combined with limited novel antifungal drugs, has promoted research on metal nanoparticles as potential alternatives [35]. Similarly, the bacteria *Escherichia coli, Pseudomonas aeruginosa,* and *Staphylococcus aureus* are among the most challenging antibiotic-resistant pathogens [36]. These opportunistic pathogens are associated with high rates of mortality and impose a great economic burden on society.

One of the algae that we have used in our work is *G. verrucosa*, common in the Mediterranean area and the Indian ocean) [37]. It is known to have antibacterial, antifungal, and antihemolytic properties [38]. Gold nanoparticles green-synthesized using *G. verrucosa* are biocompatible with normal human embryonic cells (HEK-293) [39]. However, reports are rare of the green synthesis of CuO NPs. The other seaweed that we have used is *Ulva lactuca*, also common in the northern Adriatic Sea and other temperate coasts. Its biomass contains a large amount of the polysaccharide ulvan, together with carotenoids and phenolics, and lipids and proteins, and it has abundant antioxidant activity [40].

The shallow lagoons of the northern Adriatic Sea, including Sacca di Goro, are the most important sites within the European Community for cultivation of the Manila clam (*Rudatapes philippinarum*), with a crop estimated at between 50,000 and 60,000 t y^−1^. Clam farming in Sacca di Goro is managed, mostly in a sustainable way, by cooperatives of fishermen that exploit licensed areas, under the control of regional and local authorities [41]. Clam farming has suffered serious setbacks due to massive clam mortality; uncontrolled growth of the seaweeds and the occurrence of dystrophic events triggered by decomposition of macroalgal biomass have been proposed as major factors causing the decline of clam farming in the lagoon [42]. For these reasons, the macroalgae are considered as waste and a noxious byproduct to be disposed of.

The purpose of this work was to produce useful nanoparticles from these macroalgae to reverse their role from a problem to a resource. In particular, we focused on obtaining CuO nanoparticles from an extract of the red and green macroalgae. The nanoparticles were tested on pathogenic and non-pathogenic microorganisms to determine if they have antimicrobial properties.

## 2. Results and Discussion

### 2.1. Green Synthesis of CuO Nanoparticles from Macroalgae

The copper acetate solution used for NP synthesis had initial and final pHs of 6.1 and 5.2, and 6.3 and 5.1, respectively, for the green algae (GA) and red algae (RA) extracts. The pH of the solution strongly influenced NP synthesis because changes in pH resulted in variation of their superficial charge and in their ability to bind metal cations. For gold and silver NPs, non-neutral pH increased the amount of NPs synthesized [43]. Estimates of NP size were obtained through dynamic light scattering (*dh*) and measurement of ζ-potential: 65.68 nm and −20.83 mV for green algae and 111.73 nm and −31.66 mV for red algae, respectively. Average NP dissolution has been estimated as 0.1–0.15%, which is consistent with previous analyses performed on CuO NPs [44].

From the raw extracts of both GA and RA, at the end of the microwave-assisted process the yield was 0.52 g; after pyrolysis, the produced NPs resulted in 0.15 g and 0.19 g from GA and RA, respectively. The post-pyrolysis yields of CuO NPs were 42.1% and 36.5%, based on the amount of extract, from GA and RA, respectively. There was no substantial difference in the yields of CuO NPs produced from extracts of either of the two different macroalgae.

### 2.2. XRD Nanoparticle Characterization

X-ray Powder Diffraction (PXRD) analysis resulted in equal diffractograms for the nanoparticles from the two types of algae. According to the Inorganic Crystal Structure Database (ICSD, card no. 92364), the NPs are composed of tenorite, and the dimensions could be estimated within the range of 30–40 nm for both nanoparticles deriving from red and green algae (Figure 1). The XRD diffractometry of CuO nanoparticles from *Halymenia dilatata* seaweed aqueous extract is slightly different, probably due to a capping agent formed during the nanoparticle synthesis [45].

### 2.3. TEM/EDX

The TEM images taken with Talos in Figure 2A,B showed that nanoparticles aggregate in large agglomerates in both samples, so that studying single particle dimensions and shape is quite difficult. So, the high-resolution TEM (HRTEM) technique in a JEO2200FS was used in the thinnest areas, i.e., at the edge of the agglomerates. An example of this measurement method is shown in the image in Figure 2C, where the lattice fringes of some single particles are visible, making it possible to determine their size and shape. The JEOL analyses found that the NPs are all crystalline. The aggregation dimension and the size and shape of both types of nanoparticles were quite similar, most of them being around some tens of nm (20–40) with an ovoidal/round shape. The same shape and dimension are reported in the literature for the CuO NPs obtained from *Sargassum longifolium* using a similar green synthesis procedure [46].

In samples from RA (Figure 3A), the individual crystals appear to be more agglomerated, as if they had partially ‘sintered’ (word used here simply to indicate a greater interaction between the individual particles). The diffraction patterns, reported as an inset in the A panels of Figure 3 and Figure 4, confirm that only CuO is present, in agreement with the results of XRD.

The EDX spectra from the two types of nanoparticles are shown in panel B of Figure 3 and Figure 4. The NP powders were dispersed on Ni grids for TEM observations, instead of the common Cu grids, to avoid confusing the response of the sample with that of the support. EDX measurements confirm the presence of only Cu and O in the samples; spurious signals in the spectrum represent the background of the analysis chamber.

### 2.4. FTIR Analyses

Before the heating treatment and the consequent loss of organic matter, FTIR analyses of CuO NPs were carried out to evaluate any differences in the organic functional groups derived from *U. lactuca* and *G. verrucosa.*

The two spectra are perfectly superimposable (Figure 5); the organic residues derived from the two different algae have the same composition. The FTIR spectra are insufficient to identify the organic components. However, we can hypothesize the presence of some particular functional groups, based on their known IR absorption [47,48]. The broad peak in the region 3500–3000 cm^−1^ can be seen as the result of multiple absorptions, such as those due to OH, NH, and CH stretching; the peak at 1542 cm^−1^ could suggest the presence of aromatic rings (abundant, for example, in polyphenols), as it may be attributed to the aromatic ring stretch; the peak at 1405 cm^−1^ could be due to the bending of OH bonds; the peak at 787 cm^−1^ could be a further absorption of the aromatic rings.

### 2.5. Reactive Oxygen Species (ROS) Scavenging Capacity

The DPPH assay demonstrated that there is no difference in radical scavenging capacity between the nanoparticles obtained from the macroalgae and the standard CuO NPs (Figure 6). This means that the NPs obtained from algae behaved in the same way as the standards in the presence of ROS (reactive oxygen species). Thus, CuO NPs from macroalgae are able to quench the same amount of ROS as the standard CuO NPs.

### 2.6. Microbiological Analysis on Copper Oxide (CuO) NPs

For the MIC (Minimum Inhibiting Concentration) determination, microbial cultures with OD_600_ = 0.05 were used, and serial dilutions of CuO NPs were added to the cultures. The highest CuO NP concentration under analysis was 500 mg/L, and from this 10 serial dilutions (1:2) were performed. The microorganisms treated with nanoparticles, and the untreated control, were incubated at 28 °C for 24 h; then, OD_600_ was measured. The OD_600_ values measured for each microorganism after 24 h of CuO NP treatment are reported in Table 1 and Table 2 and Appendix A. The NPs synthesized from macroalgae did not control any of the microorganisms except for *Escherichia coli*, which was inhibited by the nanoparticles from red algae at a concentration of 500 mg L^−1^. This observation is confirmed by the fact that there is a reduction in respect to the control at 250 mg L^−1^. When the microorganisms were treated with NPs from green or red algae there was a decrease in growth of at least 25% compared to the control, except for *Candida albicans*, which grew unchanged in this environment. These data, as in the case of *S. cerevisiae*, largely depend on the particle behavior in terms of stability and aggregation, and subsequent bioavailability [49,50]. In particular, Kasemets et al. [49] demonstrated how the cytotoxic effects of CuO NPs were mainly ascribed to Cu^2+^ ion release. This might depend on the medium utilized, and the dissociation rate may vary in respect to particle size and stability [44]. Flow cytometry analyses (Appendix A) on *C. albicans* and *S. cerevisiae* confirmed the different susceptibility of the two yeasts to the CuO NP treatments. Differences observed can be ascribed to differences in cell wall permeability, since standard and biosynthesized CuO NPs were highly stable. The increase of cell mortality due to the treatment, particularly for *C. albicans*, suggested that a cytotoxicity effect was observed.

From the viability test and the spot assay (Figure 7 and Appendix A), it is evident that the CuO NPs from green and red macroalgae behave in a similar way in inhibiting the growth of the microorganisms in liquid media, where bioavailability is higher. They inhibit only the growth of *Escherichia coli* and *Bacillus subtilis* but not that of the other microorganisms at any concentrations. Also, the standard NPs do not inhibit the growth of *Candida albicans* or any other microorganism. In contrast to our results, the CuO nanoparticles obtained from *Halymenia dilatata* by Sivakumar et al. [45] were most effective against *Bacillus subtilis*. Thus, the different types of algae used to prepare the CuO NPs have an influence on their biological properties. Similarly, Zhang et al. [51] reported that CuO NPs obtained by green synthesis were found to have a potential antibacterial effect against *E. coli*; they did not screen other microorganisms. No inhibitory effects on growth were observed in solid media. This result can be ascribed to the limited dissolution of the CuO NPs utilized and to their limited bioavailability in solid media.

## 3. Materials and Methods

### 3.1. Macroalgae Biomass Collection and Extraction Process

Biomass from green (*Ulva lactuca* L.) and red (*Gracilaria verrucosa* (H.) Papenfuss) macroalgae was collected in Sacca di Goro (Italy) in October 2023. Detailed descriptions of the two types of macroalgae utilized are reported in Dominguez and Loret [52] and Fredericq and Hommersand [53], respectively. Green and red macroalgae biomass was washed at the site with seawater and immediately brought to the laboratory, where it was manually washed with tap water, then divided to obtain two different samples of both green and red biomass. The macroalgae were finally washed again three times with distilled water.

The biomass was dried at room temperature for seven days (Appendix A) and ground using a Kenwood Chopper CHP61. Macroalgae extraction was performed using the methods of Fatima et al. [54] and Jayarambabu et al. [55]. Macroalgae extract was obtained with a biomass/ethanol 1:30 ratio on a hot-plate at 70 °C, under magnetic stirring (750 rpm) for 4 h. At the end, sonication was performed using a Scientific Model 505 Sonic Dismembrator (Fisher Scientific, Waltham, MA, USA) at 40% amplitude for 60 s to maximize dispersion. The extract was collected using centrifugation at 13,000 rpm for 5 min at 4 °C and stored as an ethanol solution at 4 °C before NP synthesis.

### 3.2. NP Synthesis

Cu-NP synthesis was carried out following a modified version of the methodology of Jayarambabu et al. [55]. Copper acetate (Merck, Darmstadt, Germany) solution 0.1 M was added to macroalgae extract at a 1:2 *v*/*v* ratio. Synthesis was performed by a microwave-assisted method. The mixed solution of copper acetate and macroalgae extract was kept for 30 s in a microwave oven at 600 W power; this operation was repeated 8 times. The synthesis of raw Cu-NPs was completed when the color of the solution changed from deep blue to light greenish blue (Appendix A). Isolation of raw NPs was performed using centrifugation at 13,000 rpm for 5 min at 4 °C; the pellet was dried at room temperature overnight. This pellet (Appendix A) was composed of copper oxide NPs and a precipitate of algal extract organic compounds.

The pellet was transferred into glass vials and dried at 80 °C in a vacuum overnight to remove excess water. The temperature was then increased to 110 °C and maintained for 2 h to remove final traces of water. The dry weight of the sample was noted, and the FTIR spectra were acquired before the heat treatment to decompose the organic matrix.

In the case of small batches, the sample was transferred into alumina vials and treated at 550 °C for 5 h in a tubular furnace and oxygen/nitrogen atmosphere at a ratio of 1:5. For larger batches, a muffle mold was used, heated to the same temperature but performing the incineration in air. Interestingly, there were no appreciable differences using either approach for heat treatment or variation of the heating curves. The samples were then collected and re-weighed to determine the weight loss due to the thermal treatment. CuO nanoparticles were obtained from both green and red macroalgae, as verified with XRD and HRTEM/EDX.

### 3.3. Nanoparticle Characterization

#### 3.3.1. XRD

X-ray Powder Diffraction (PXRD) analysis was performed to assess the structural properties of the obtained CuO NPs using a Rigaku Smartlab XE diffractometer in Bragg–Brentano geometry with Cu Kα wavelength (λ  =  1.5406 Å) and a Ni filter to suppress the Kβ contribution; 5.0° Soller slits were used both on the incident and diffracted beam, and data were collected using a HyPix3000 detector. Measurements were performed in the 10–80° 2θ range with a 0.05 step size, acquired in continuous 1D mode (2° min^−1^).

All the observed reflections were indexed as belonging to monoclinic CuO (tenorite), according to the Inorganic Crystal Structure Database (ICSD, card no. 92364). Mean crystal size dimensions were calculated by pseudo-Voight fit and estimated to be within the 30 to 40 nm range. Notably, no spurious phases were observed (Figure 1).

#### 3.3.2. HRTEM/EDX

We utilized initially a Talos high-resolution TEM (Talos F200S G2, SEM FEG, Thermo Fisher Scientific, Waltham, MA, USA) equipped with an EDX detector. Nanoparticles were sonicated in Eppendorf tubes containing ethanol for 20 min; they were then spotted on Au TEM grids and allowed to evaporate for 15 min at room temperature. We observed the morphology of single particles (Figure 2) and of aggregates and took the EDX spectra at 80 KeV (Figure 2).

Then, we verified the results obtained with the Talos TEM using a JEOL JEM2200FX (Tokyo, Japan) field emission analytical Scanning TEM microscope operated at 200 kV, equipped with two high-angle annular dark field detectors for Z contrast detection, a built-in W filter for electron energy loss spectroscopy, and an Oxford Aztec Energy TEM EDX system mounting the XPLORE Silicon Drift Detector with an 80 mm^2^ active area. The samples were dispersed on Nichel grids covered by continuous ultrathin carbon film at room temperature (Figure 3 and Figure 4).

#### 3.3.3. Zeta Potential, Particle Size and Dissolution

The average particle size (*dh*) and zeta (ζ) potential of the nanomaterials (100 mg L^–1^) were determined in ddH_2_O (double distilled water) on a Zetasizer Nano Series ZS90 (Malvern Instruments, Malvern, UK), as described in Pagano et al. [56]. For particle suspension, CuO NPs for green and red algae (100 mg L^−1^) were prepared in ddH_2_O. Samples were collected after 1, 5, and 10 d. Aliquots of each sample (1 mL) were precipitated by ultracentrifugation at 30,000 rpm for 10 min at 20 °C (Optima Max-XP ultracentrifuge, Beckman-Coulter Inc., Brea, CA, USA). The liquid phase was collected and digested in 4 mL of 1 M HNO_3_ (67% *w*/*w*) for 40 min at 200 °C using a VELP DK20 digester (VELP Scientifica, Usmate, Italy). Analysis was performed by flame atomic absorption spectroscopy (FA-AAS; AA240FS, Agilent Technologies, Santa Clara, CA, USA) for the presence of Cu (324.7 nm), as in Marmiroli et al. (2021) [44].

#### 3.3.4. FTIR

Infrared (IR) spectra were recorded with an Agilent Cary 630 FTIR spectrophotometer equipped with the attenuated total reflection (ATR) accessory (diamond), range 4000–650 cm^−1^. For each sample, a small aliquot of dry powder was placed on the ATR system and tamped down, and then the FTIR spectrum was acquired directly (Figure 5).

#### 3.3.5. DPPH Assay for Free Radical Scavenging Capacity

The 2,2-Diphenyl-1-picrylhydrazyl assay (DPPH assay) is a colorimetric method utilized to measure the radical scavenging activity of antioxidant compounds. According to Pagano et al. [56], an aliquot of 50 μL of plant extract was added to 1.95 mL of DPPH solution 0.06 mM in methanol; after 30 min at ambient temperature, the absorbance at 520 nm was measured (Varian Cary 50 spectrophotometer, Agilent Technologies). The absorbance was also read after 40 min and 50 min of incubation, to verify that a steady state was achieved (plateau in the curve). Appropriate solvent blanks were run in each assay (Figure 6).

#### 3.3.6. Microbiological Analysis on Copper Oxide (CuO) Nanoparticles (NPs)

Five microorganisms were used: *Saccharomyces cerevisiae* BY4742, *Candida albicans* SC5314, *Escherichia coli* DH5α, *Bacillus subtilis* BV84, and *Staphylococcus aureus* ATCC 6538. *C. albicans* and *S. cerevisiae* were grown in YPD medium (yeast extract 1%, peptone 2%, and dextrose 2%), while *E. coli*, *B. subtilis*, and *S. aureus* were grown in Luria Bertani (LB) medium (yeast extract 0.5%, tryptone 1%, and NaCl 1%). For each strain, a single colony was isolated from agar medium and cultured in LB and YPD broth for 24 h at 28 °C, with agitation. Then, OD_600_ was measured, and Minimum Inhibitory Concentration (MIC) determination was performed. Three different CuO NPs were used: standard (STD) NPs purchased from US Research Nanomaterials, Inc. (Houston, TX, USA) and previously characterized by Marmiroli et al. [44], together with NPs synthesized from green algae and NPs synthesized from red algae.

For the MIC determination, microbial cultures with OD_600_ 0.05 were used, and serial dilutions of CuO NPs were added to the cultures. The spot assay was also performed in solid media: serial dilutions of microorganisms treated with CuO NPs on solid media (1 OD, serial dilutions 1:10) were performed; 10 µL of cell culture was plated on agar medium after 24 h of incubation at 28 °C with the NP treatments under analysis, as reported in Appendix A. The highest CuO NP concentration under analysis was 500 mg L^−1^, and a further 10 serial dilutions (1:2) were performed. The microorganisms treated with nanoparticles, and the untreated control, were incubated at 28 °C for 24 h; then, OD_600_ was measured. The OD_600_ values obtained for each microorganism after 24 h of CuO NP treatment are reported in Table 1 and Table 2 and Appendix A. To better assess cell viability and the potential nanoparticle cytotoxic effects on the microorganisms under analysis, 10 μL of culture obtained from the three highest nanoparticle concentrations (500 mg L^−1^, 250 mg L^−1^, and 125 mg L^−1^) was plated on Petri dishes at 6 different cellular dilutions (10^−3^–10^−8^). The images of the spots obtained after 24 h of incubation at 28 °C are reported in Table 1 and Table 2 and Figure 7.

#### 3.3.7. Flow Cytometry on Yeasts

*S. cerevisiae* and *C. albicans* strains were grown on liquid YPD for 24 h at 28 °C in the absence of NPs as control or in the presence of 250 mg L^−1^ of each type of nanomaterial used (Standard, or green or red algae product). A total of 10^7^ cells per treatment were treated with propidium iodide (PI) as a vital stain (5 μg mL^−1^) for 20 min at 20 °C. As an internal standard, a sample was included with induced 50% cell mortality through thermal treatment at 100 °C for 20 min. Samples were analyzed with an Attune NxT Flow Cytometer (Thermo Fisher Scientific, Wyman Street, Waltham, MA, USA). Data were analyzed with Attune NxT software v 3.1. A graph of the signal derived from PI analysis of cell mortality is presented in Appendix A.

## 4. Conclusions

Macroalgae, especially *Ulva lactuca* and *Gracilaria verrucosa*, are considered as invading and noxious species by clam fishermen in the northern Adriatic Sea in Italy. Every year, this results in tons of macroalgae biomass being left to rot on the sand, creating a foul odor. This causes problems for the clam fishermen and is also detrimental for tourism in the area. We have demonstrated that it is possible with green synthesis methods to obtain CuO nanoparticles of diameter 20–50 nm from biomass of these species. These nanoparticles are valuable because of their antimicrobial potential. Thus, a marine nuisance can be converted to a useful material, which may be exploited for human wellbeing.

## Figures and Tables

**Figure 1 nanomaterials-14-01157-f001:**
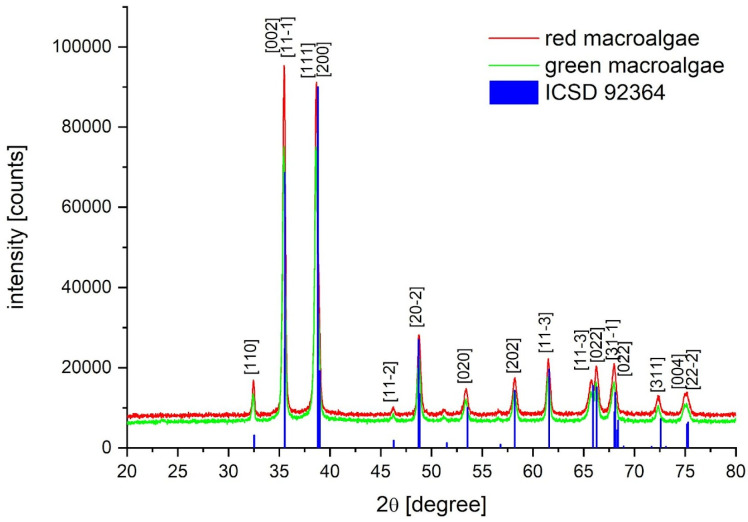
Diffractograms of the nanoparticles produced from the two macroalgae.

**Figure 2 nanomaterials-14-01157-f002:**
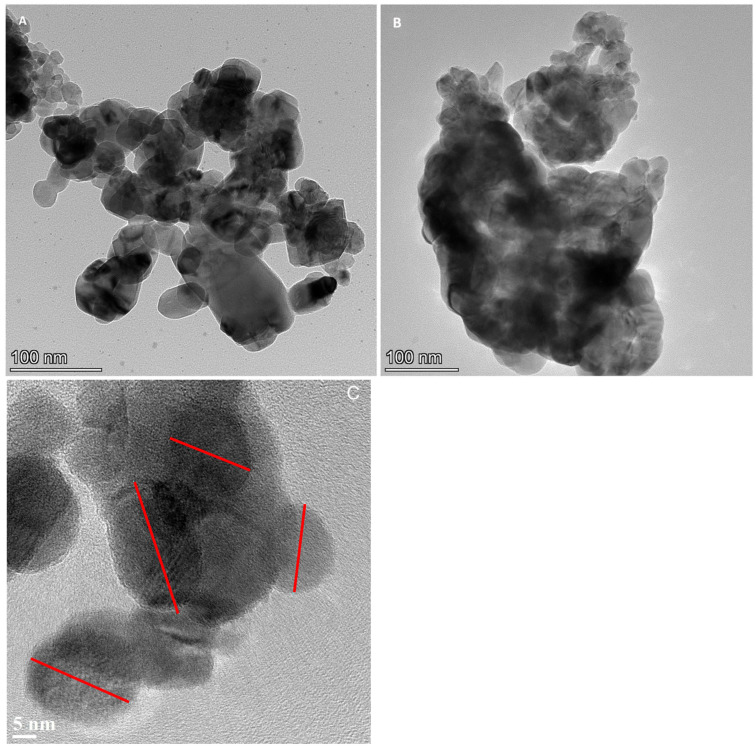
TEM images of nanoparticles from (**A**) green algae (*U. lactuca*) and (**B**) red algae (*G. verrucosa*). In (**C**), a typical HRTEM image of green algae is shown, where the size, shape, and dimension of some individual particles can be seen (red bars).

**Figure 3 nanomaterials-14-01157-f003:**
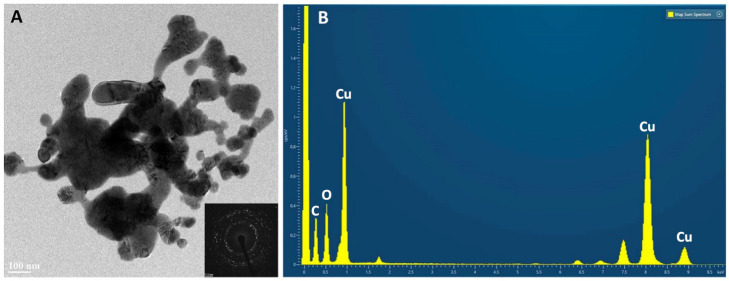
Nanoparticles from *G. verrucosa*. (**A**) TEM, with the diffraction pattern as inset. (**B**) EDX spectrum.

**Figure 4 nanomaterials-14-01157-f004:**
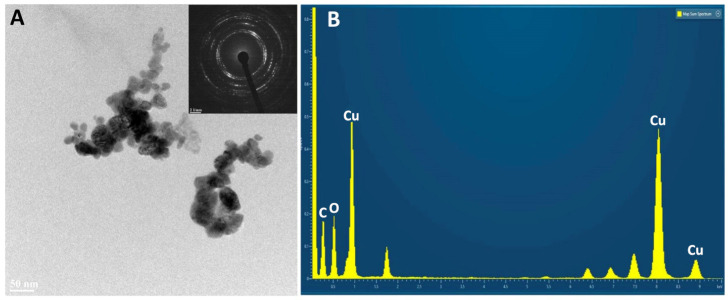
Nanoparticles from *U. lactuca*. (**A**) TEM, with the diffraction pattern as inset. (**B**) EDX spectrum.

**Figure 5 nanomaterials-14-01157-f005:**
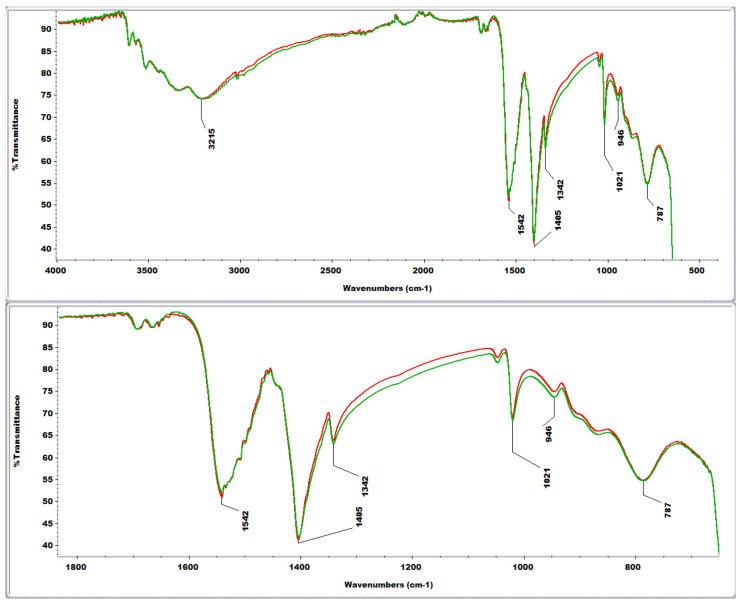
FTIR spectra of the nanoparticles derived from *U. lactuca* (green) and *G. verrucosa* (red), before the final heat treatment.

**Figure 6 nanomaterials-14-01157-f006:**
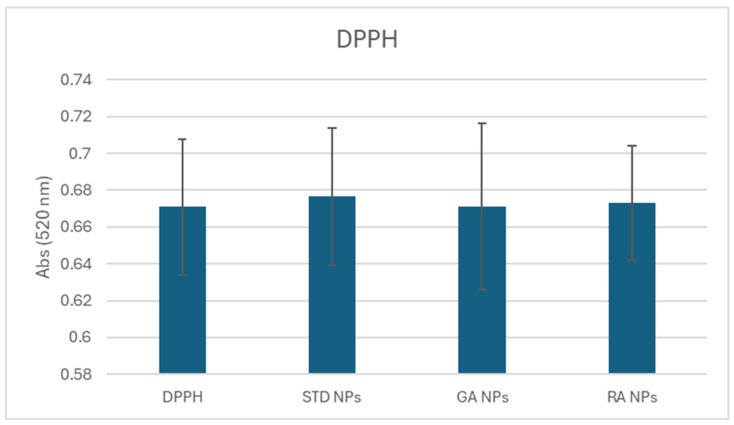
DPPH results for the nanoparticles from GA NPs and RA NPs in comparison with standard CuO NPs.

**Figure 7 nanomaterials-14-01157-f007:**
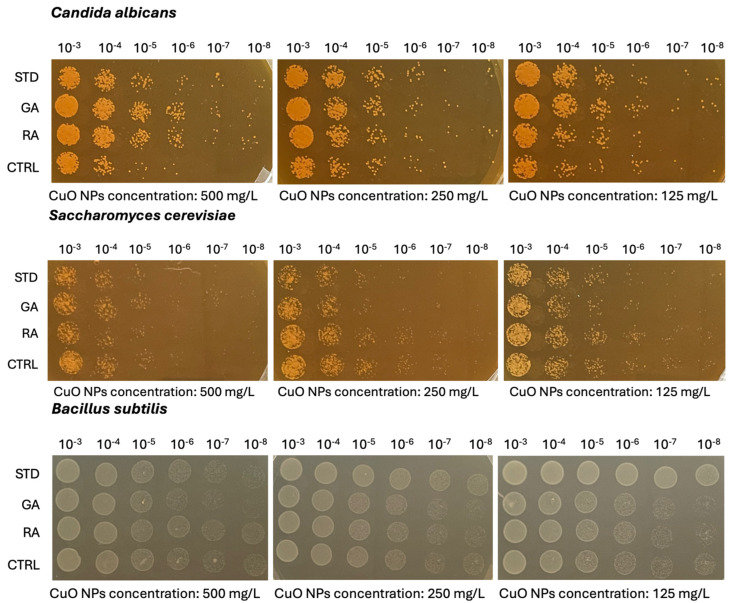
Viability test of serial dilutions (from 10^−3^ to 10^−8^ cells mL^−1^) of microorganisms treated in liquid media with CuO NPs. STD: standard CuO NPs; GA: CuO NPs synthesized from green algae; RA: CuO NPs synthesized from red algae; CTRL: untreated microorganisms. Ten µL of cell culture was plated on agar media after 24 h of incubation at 28 °C with the various NPs.

**Table 1 nanomaterials-14-01157-t001:** Growth inhibition by NPs. STD, commercial product; GA, green algae NPs; RA, red algae NPs. A graphical representation is given in Appendix A.

	** *Candida albicans* **											
	**NP Concentration (mg L^−1^)**											
	**500**	**250**	**125**	**62.5**	**31.3**	**15.6**	**7.8**	**3.9**	**2.0**	**1.0**	**0.5**	**CTRL**
STD NPs	1.098	1.233	1.31	1.342	1.349	1.335	1.333	1.361	1.364	1.377	1.372	1.341
GA NPs	1.092	1.259	1.326	1.363	1.362	1.367	1.386	1.393	1.382	1.383	1.391	1.343
RA NPs	1.11	1.304	1.363	1.383	1.374	1.398	1.407	1.387	1.399	1.393	1.396	1.407
	** *Saccharomyces cerevisiae* **											
	**NP Concentration (mg L^−1^)**											
	**500**	**250**	**125**	**62.5**	**31.3**	**15.6**	**7.8**	**3.9**	**2.0**	**1.0**	**0.5**	**CTRL**
STD NPs	0.915	1.021	1.111	1.13	1.218	1.141	1.246	1.249	1.248	1.249	1.206	1.114
GA NPs	0.863	0.982	1.079	1.177	1.158	1.191	1.226	1.253	1.235	1.283	1.314	1.231
RA NPs	0.831	1.013	1.129	1.127	1.2	1.233	1.248	1.304	1.304	1.334	1.375	1.354
	** *Bacillus subtilis* **											
	**NP Concentration (mg L^−1^)**											
	**500**	**250**	**125**	**62.5**	**31.3**	**15.6**	**7.8**	**3.9**	**2.0**	**1.0**	**0.5**	**CTRL**
STD NPs	0.05	0.05	0.05	0.11	0.634	0.921	0.965	0.996	0.965	1.025	0.995	0.997
GA NPs	0.566	0.856	0.921	0.985	1.005	1.025	1.008	1.013	1.007	0.966	0.975	0.999
RA NPs	0.613	0.811	0.892	0.964	0.984	1.014	1.037	0.998	1.029	1.028	1.022	1.053
	** *Staphylococcus aureus* **											
	**NP Concentration (mg L^−1^)**											
	**500**	**250**	**125**	**62.5**	**31.3**	**15.6**	**7.8**	**3.9**	**2.0**	**1.0**	**0.5**	**CTRL**
STD NPs	0.619	0.549	0.618	0.596	0.536	0.561	0.563	0.529	0.55	0.588	0.6	0.675
GA NPs	0.481	0.473	0.555	0.523	0.528	0.542	0.534	0.523	0.54	0.552	0.538	0.666
RA NPs	0.455	0.371	0.542	0.452	0.531	0.607	0.506	0.549	0.465	0.58	0.52	0.69

**Table 2 nanomaterials-14-01157-t002:** Summary of the effects of the nanoparticles on microorganisms. STD, commercial product; GA, green algae NPs; RA, red algae NPs; nd, undetermined.

**Growth Inhibition (OD_600_ = 0.05)**			
**NP concentrations (mg L^−1^)**			
	STD NPs	GA NPs	RA NPs
*Candida albicans*	nd	nd	nd
*Saccharomyces cerevisiae*	nd	nd	nd
*Bacillus subtilis*	125	nd	nd
*Escherichia coli*	nd	nd	500
*Staphylococcus aureus*	nd	nd	nd
**50% growth reduction compared to the untreated control**			
**NP concentrations (mg L^−1^)**			
	STD NPs	GA NPs	RA NPs
*Candida albicans*	nd	nd	nd
*Saccharomyces cerevisiae*	nd	nd	nd
*Bacillus subtilis*	62.5	nd	nd
*Escherichia coli*	nd	nd	250
*Staphylococcus aureus*	nd	nd	nd
**25% growth reduction compared to the untreated control**			
**NP concentrations (mg L^−1^)**			
	STD NPs	GA NPs	RA NPs
*Candida albicans*	nd	nd	nd
*Saccharomyces cerevisiae*	nd	500	250
*Bacillus subtilis*	31.25	500	500
*Escherichia coli*	nd	500	125
*Staphylococcus aureus*	nd	500	250

## Data Availability

Data is contained within the article and Appendix A.

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
