# Peer review of "Green Synthesis of CuO Nanoparticles from Macroalgae Ulva lactuca and Gracilaria verrucosa"

_nanomaterials, 2024, doi:10.3390/nano14131157_

Round 1

Reviewer 1 Report

Comments and Suggestions for Authors

Dear authors,

I would like to acknowledge the nice idea to solve the problem of macroalgae overabundance in the field of clam culture, by transforming a waste into a useful product, in this case, biosynthesizing CuO nanoparticles (NPs).

The manuscript is well written, clearly explaining the aim of the study and the required techniques to verify the antimicrobial utility of the algae, and also to explain it.

However, I found it very difficult to understand several aspects of the study, maybe this might be clarified or better explained:

1.      Why/How biosynthesized NPs might acquire antimicrobial properties?

2.      I could not understand why the IR is performed with NPs still displaying organic matter from the algae, before the final heat treatment (as these will obviously show the presence of organic matter that might be associated with organic molecules displaying antioxidant and antimicrobial properties), if the microbiological analysis are performed with heat-treated NPs that would not exhibit this organic matter anymore.

3.      Within the microbiological analysis, it is claimed that the bioavailability in liquid media is higher (line 248), but according to the authors, the spot assay only shows inhibition of the growth of E. Coli, but not that of the other microorganisms at any concentrations (line 248-249)… Besides, even if I am not an expert, by simple observation of spot assays I would say the growth of Bacilus subtilis diminishes for green algae NPs (500 mg/L) more clearly than E. Coli, and same stands for Saccharomyces with green algae NPs (250 mg/L).

Besides clarifying these questions, there are some suggestions that might improve the manuscript:

1.      Within the introduction, line 30, it is claimed that NPs display unique properties due to their high surface area to volume ratio, and it is true, but confinement effects are also responsible, so this should be included within the text. This references might help the reader verify this:

a.      Ortiz de Zárate, D.; García-Meca, C.; Pinilla-Cienfuegos, E.; Ayúcar, J.A.; Griol, A.; Bellières, L.; Hontañón, E.; Kruis, F.E.; Martí, J. Green and Sustainable Manufacture of Ultrapure Engineered Nanomaterials. Nanomaterials 2020, 10, 466. https://doi.org/10.3390/nano10030466

b.      Carenco, S.; Portehault, D.; Boissière, C.; Mézailles, N.; Sanchez, C. 25th Anniversary Article: Exploring Nanoscaled Matter from Speciation to Phase Diagrams: Metal Phosphide Nanoparticles as a Case of Study. Adv. Mater. 2014, 26, 371–390. https://doi.org/10.1002/adma.201303198

2.      Line 42, the word “approach” might be change by a synonym to avoid repetition.

3.      Line 173 makes reference to x-ray spectra, while it would be better to use the EDX analysis.

4.      Figure 2 does not make reference to which algae is the red or the green one (the reference used in the discussion), and the reader might have forgotten it. It might be helpful to indicate this information within the caption.

5.      Line 343, ddH2O stands for double distilled water, I assume. This should be explained.

6.      Line 381, spot assay. I do not understand if the dilutions have been performed with NPs and also pathogens, or only pathogens. If NPs have also been diluted, then the concentration is not the same… Besides, the control of Staphylococcus aureus does not show any reduction with dilution.

Author Response

Reviewer 1

Dear authors,

I would like to acknowledge the nice idea to solve the problem of macroalgae overabundance in the field of clam culture, by transforming a waste into a useful product, in this case, biosynthesizing CuO nanoparticles (NPs).

The manuscript is well written, clearly explaining the aim of the study and the required techniques to verify the antimicrobial utility of the algae, and also to explain it.

We would like to thank the Rev1 for his positive comments on our work.

However, I found it very difficult to understand several aspects of the study, maybe this might be clarified or better explained:

  1. Why/How biosynthesized NPs might acquire antimicrobial properties?

All types of nanoparticles, especially metal-based NPs show antimicrobial properties. Therefore, it has been supposed and proved that also biosynthesized metal nanoparticles can show antimicrobial properties. This can be ascribed to different physico-chemical properties such as stability (and consequently metal release), shape, size, charge. In addition, the presence on the nanoparticles of capping agents coming from the natural compounds may enhance their biocidal properties (Rajeskumar et al, 2021).

  1. I could not understand why the IR is performed with NPs still displaying organic matter from the algae, before the final heat treatment (as these will obviously show the presence of organic matter that might be associated with organic molecules displaying antioxidant and antimicrobial properties), if the microbiological analysis are performed with heat-treated NPs that would not exhibit this organic matter anymore.

The IR analyses were performed after the syntheses to evaluate if the residual organic matter was different for the two algae or not; the sense of the analyses was to have a comparison between the two syntheses when the organic residue was still present on the nanoparticles. After the final heat treatment, as you properly highlight, the organic residue is lost and it’s not possible to analyze it anymore.

The IR analyses were not enough to elucidate the composition of the residue, as written in the paper it is only possible to formulate an hypothesis about the presence of some functional groups; however, the main information was that the products derived from the two algae have the same organic residue. After the heat treatment such information would be lost.

The paragraph has been modified as reported.

 Lines 179-181:

“Before the heating treatment and the consequent loss of organic matter, FTIR analyses of CuO NPs were carried out to evaluate any differences in the organic functional groups derived from U. lactuca and G. verrucosa.”

  1. Within the microbiological analysis, it is claimed that the bioavailability in liquid media is higher (line 248), but according to the authors, the spot assay only shows inhibition of the growth of E. Coli, but not that of the other microorganisms at any concentrations (line 248-249)… Besides, even if I am not an expert, by simple observation of spot assays I would say the growth of Bacilus subtilis diminishes for green algae NPs (500 mg/L) more clearly than E. Coli, and same stands for Saccharomyces with green algae NPs (250 mg/L).

The text has been modified to improve the clarity of the results:

Lines 227-232:

“From the viability test and the spot assay (Figures 7 and S2) it is evident that the CuO NPs from Green and Red macroalgae behave in a similar way in inhibiting the growth of the microorganisms in liquid media, where the bioavailability is higher. They inhibit only the growth of Escherichia coli and Bacillus subtillis but not that of the other micro-organisms at any concentrations. Also, the standard NPs do not inhibit the growth of Candida albicans or any other microorganism.”

Lines 351-376: 

“Five microorganisms were used: Saccharomyces cerevisiae BY4742, Candida albicans SC5314, Escherichia coli DH5α, Bacillus subtilis BV84, and Staphylococcus aureus ATCC 6538. C. albicans and S. cerevisiae were grown in YPD medium (yeast extract 1%, peptone 2%, dextrose 2%), while E. coli, B. subtilis and S. aureus were grown in Luria Bertani (LB) medium (yeast extract 0.5%, tryptone 1%, NaCl 1%). For each strain, a single colony was isolated from agar medium and cultured in LB and YPD broth for 24 h at 28 °C, with agitation. Then, OD600 was measured, and Minimal Inhibitory Concentration (MIC) de-termination was performed. Three different CuO NPs were used: standard (STD) NPs purchased from US Research Nanomaterials, Inc. (Houston, TX, USA) and previously characterized by Marmiroli et al. [44], together with NPs synthesized from green algae and NPs synthesized from red algae. For the MIC determination, microbial cultures with OD600 0.05 were used and serial dilutions of CuO NPs were added to the cultures. The spot assay been also performed in solid media: serial dilution of microorganisms treated with CuO NPs on solid media (1 OD, serial dilutions 1:10 have been performed; 10 µL of cell culture were plated on agar medium after 24 h of incubation at 28 °C with the NPs treatments under analysis, as reported in Figure S2. The highest CuO NPs concentration under analysis was 500 mg L-1 and a further 10 serial dilutions (1:2) were performed. The microorganisms treated with nanoparticles, and the untreated control, were incubated at 28 °C for 24 h; then the OD600 was measured. The OD600 values obtained for each microorganism after 24 h of CuO NPs treatment are reported in Tables 1-2 and Figure S1. To better assess cell viability and the potential nanoparticle cytotoxic effects on the microorganisms under analysis, 10 uL of culture obtained from the three highest nanoparticles concentrations (500 mg L-1, 250 mg L-1, 125 mg L-1) were plated on Petri dishes at 6 different cellular dilutions (10-3-10-8). The images of the spots obtained after 24 h of incubation at 28 °C are reported in Tables 1-2 and Figures 7.”

Figure 7 caption has been also modified accordingly: “Viability test of serial dilutions (from 10-3 to 10-8 cells mL-1) of microorganisms treated in liquid media with CuO NPs. STD: standard CuO NPs, GA: CuO NPs synthesized from green algae, RA: CuO NPs synthesized from red algae, CTRL: untreated microorganisms. Ten µL of cell culture were plated on agar media after 24 h of incubation at 28 °C with the various NPs.”

Besides clarifying these questions, there are some suggestions that might improve the manuscript:

  1. Within the introduction, line 30, it is claimed that NPs display unique properties due to their high surface area to volume ratio, and it is true, but confinement effects are also responsible, so this should be included within the text. This references might help the reader verify this:

  1. Ortiz de Zárate, D.; García-Meca, C.; Pinilla-Cienfuegos, E.; Ayúcar, J.A.; Griol, A.; Bellières, L.; Hontañón, E.; Kruis, F.E.; Martí, J. Green and Sustainable Manufacture of Ultrapure Engineered Nanomaterials. Nanomaterials 2020, 10, 466. https://doi.org/10.3390/nano10030466

  1. Carenco, S.; Portehault, D.; Boissière, C.; Mézailles, N.; Sanchez, C. 25th Anniversary Article: Exploring Nanoscaled Matter from Speciation to Phase Diagrams: Metal Phosphide Nanoparticles as a Case of Study. Adv. Mater. 2014, 26, 371–390. https://doi.org/10.1002/adma.201303198.

The references have been included in the introduction as refs. 5-6.

  1. Line 42, the word “approach” might be change by a synonym to avoid repetition.

The sentence has been modified as reported: “NPs can be synthesized by two fundamental methods: top-down and bottom-up.”

  1. Line 173 makes reference to x-ray spectra, while it would be better to use the EDX analysis.

The sentence has been modified as reported: “The EDX spectra from the two types of nanoparticles are shown in panels B of Figure 3 and 4. The NP powders were dispersed on Ni grids for TEM observations, instead of common Cu grids, to avoid confusing the response of the sample with that of the support. EDX measurements confirm the presence of only Cu and O in the samples; spurious signals in the spectrum represent the background of the analysis chamber.”

  1. Figure 2 does not make reference to which algae is the red or the green one (the reference used in the discussion), and the reader might have forgotten it. It might be helpful to indicate this information within the caption.

Thank you for the comment. The Fig 2 caption has been updated: “Figure 2. Nanoparticles from (A) green algae (U. lactuca) and (B) red algae (G. verrucosa).”

  1. Line 343, ddH2O stands for double distilled water, I assume. This should be explained.

It has been explained that ddH2O is double distilled water.

  1. Line 381, spot assay. I do not understand if the dilutions have been performed with NPs and also pathogens, or only pathogens. If NPs have also been diluted, then the concentration is not the same… Besides, the control of Staphylococcus aureus does not show any reduction with dilution.

We performed both assays: in the first case, reported in Figure 7, thought viability test we exposed the microorganisms to different concentration of NPs (standard, from green synthesis both from green and red algae). After 24h treatment in liquid media a serial dilution of the strains has been spotted on solid agar to assess the cell viability.

In the second case, reported in supplementary (Figure S2), to assess the inhibition concentration for the growth. The spot assay has been performed with serial dilution of the strains directly on solid agar medium, that contained different concentration of nanoparticles.

To clarify these points the methods section 3.3.6 has been edited as reported:

Lines 351-376:

“Five microorganisms were used: Saccharomyces cerevisiae BY4742, Candida albicans SC5314, Escherichia coli DH5α, Bacillus subtilis BV84, and Staphylococcus aureus ATCC 6538. C. albicans and S. cerevisiae were grown in YPD medium (yeast extract 1%, peptone 2%, dextrose 2%), while E. coli, B. subtilis and S. aureus were grown in Luria Bertani (LB) medium (yeast extract 0.5%, tryptone 1%, NaCl 1%). For each strain, a single colony was isolated from agar medium and cultured in LB and YPD broth for 24 h at 28 °C, with agitation. Then, OD600 was measured, and Minimal Inhibitory Concentration (MIC) de-termination was performed. Three different CuO NPs were used: standard (STD) NPs purchased from US Research Nanomaterials, Inc. (Houston, TX, USA) and previously characterized by Marmiroli et al. [44], together with NPs synthesized from green algae and NPs synthesized from red algae.

For the MIC determination, microbial cultures with OD600 0.05 were used and serial dilutions of CuO NPs were added to the cultures. The spot assay been also performed in solid media: serial dilution of microorganisms treated with CuO NPs on solid media (1 OD, serial dilutions 1:10 have been performed; 10 µL of cell culture were plated on agar medium after 24 h of incubation at 28 °C with the NPs treatments under analysis, as reported in Figure S2. The highest CuO NPs concentration under analysis was 500 mg L-1 and a further 10 serial dilutions (1:2) were performed. The microorganisms treated with nanoparticles, and the untreated control, were incubated at 28 °C for 24 h; then the OD600 was measured. The OD600 values obtained for each microorganism after 24 h of CuO NPs treatment are reported in Tables 1-2 and Figure S1. To better assess cell viability and the potential nanoparticle cytotoxic effects on the microorganisms under analysis, 10 uL of culture obtained from the three highest nanoparticles concentrations (500 mg L-1, 250 mg L-1, 125 mg L-1) were plated on Petri dishes at 6 different cellular dilutions (10-3-10-8). The images of the spots obtained after 24 h of incubation at 28 °C are reported in Tables 1-2 and Figures 7.”

Regarding S. aureus, as reported in the text, no significant changes in viability have been observed in solid media, while in liquid media the MIC has been estimated for green and red algae as 500 and 250 mg L, respectively. Table 2 has been modified to give more relevance to the point.

Figure 7 caption has been also modified accordingly: “Viability test of serial dilutions (from 10-3 to 10-8 cells mL-1) of microorganisms treated in liquid media with CuO NPs. STD: standard CuO NPs, GA: CuO NPs synthesized from green algae, RA: CuO NPs synthesized from red algae, CTRL: untreated microorganisms. Ten µL of cell culture were plated on agar media after 24 h of incubation at 28 °C with the various NPs.”

Reviewer 2 Report

Comments and Suggestions for Authors

1.       Removal of underlinings from lines 76, 360 and bibliographical references from lines 92, 116, 117 and, 119.

2.       The reference to Rajeshkumar et al. 2021 on line 173 should be updated to refer to the article with DOI: 10.1016/j.molstruc.2021.130724.

3.       Line 145: The term "increase synthesis" should be clarified.

4.       Line 149: Specify the solvent for NP dissolution.

5.       Line 161, how can you determine the size of NPs via PXRD analysis?

6.       Lines 170-171, through which software/procedure can you determine the average size of the NPs? And their shape? The HR-TEM images (Figure 2) do not look sharp enough to say this.

7.       Lines 185-186: Clarify whether the deposition of the sample for characterization was done at room temperature or higher temperatures.

Author Response

Reviewer 2

Removal of underlinings from lines 76, 360 and bibliographical references from lines 92, 116, 117 and, 119.

The underlined words and bibliography have been removed.

  1. The reference to Rajeshkumar et al. 2021 on line 173 should be updated to refer to the article with DOI: 10.1016/j.molstruc.2021.130724.

In line 173 we intended to use as citation the article of Rajeskumar et al, Med Cell Longev. 2021, 2021, not the one with the doi that the Reviewer wants to be used. The citatation of Rejeskumar et al., 2021 with DOI: 10.1016/j.molstruc.2021.130724 has been included in the introduction (ref. 20).

Rajeshkumar S, Nandhini NT, Manjunath K, Sivaperumal P, Prasad GK, Alotaibi SS, Roopan SM. Environment friendly synthesis copper oxide nanoparticles and its antioxidant, antibacterial activities using Seaweed (Sargassum longifolium) extract. Journal of Molecular Structure, 2021, 1242, 130724.

  1. Line 145: The term "increase synthesis" should be clarified.

The sentence has been clarified in “The amount of NPs Synthetized”.

  1. Line 149: Specify the solvent for NP dissolution.

As explained in paragraph 3.3.2., HR TEM/EDX, the solvent was ethanol.

  1. Line 161, how can you determine the size of NPs via PXRD analysis?

Determining the size of nanoparticles using PXRD analysis is a standard procedure in material science and involves using the so-called Scherrer equation. This experimental law allows for just an estimation of the crystallite size, basing on the broadening of the diffraction peaks: such effect is due, among other contributions, to the non-infinite (rather small, actually) nature of the crystal under investigation and the NP size is proportional to 1/FWHM of the peaks. From a more fundamental point of view, the atomic scattering factor can be considered as the Fourier transform of the electron density distribution in the solid: if a crystal of finite size is considered, the electron density function is truncated, which corresponds to a convolution in reciprocal space, and this results in the broadening of the diffraction peaks.

  1. Lines 170-171, through which software/procedure can you determine the average size of the NPs? And their shape? The HR-TEM images (Figure 2) do not look sharp enough to say this.

The NP average size has been estimated using a software provided with the PXRD apparatus in use (Rigaku Smartlab XE suite) and is based on the Scherrer equation (as reported in answer to query n.5).

We thank the reviewer for the observation about the particle shape. It was our mistake to indicate the images in figure 2 as HRTEM, since the two images are not. The shape of the particles can be usually studied using TEM, but in this case it was not possible due to their very strong tendency to agglomerate (Fig 2A, B). So, the high-resolution TEM (HRTEM) technique was used in the thinnest areas, i.e. at the edge of the agglomerates. The example of this measurement method is shown in the new photo in figure 2c, added to the figure of the previous version.

In this kind of images, as shown in figure 2c, it is possible to isolate the lattice fringes of each individual particle and determine its shape and size.

 TEM analysis is a "local" investigation compared with PXRD, which instead is a large scale one. Nevertheless, it is possible to compare the results of the two techniques by taking several HRTEM pictures from different areas. There is good agreement between the results of the two techniques, considering the limitations of both.

The entire paragraph 2.3 TEM-EDX has been amended as follows to make it clearer and meet the referee’s requests:

2.3. TEM/EDX

The TEM images taken with Talos in Figure 2A, B showed that nanoparticles aggregate in large agglomerates in both samples so that studying single particles dimensions and shape is quite difficult. So, the high-resolution TEM (HRTEM) technique in a JEO2200FS was used in the thinnest areas, i.e. at the edge of the agglomerates. An example of this measurement method is shown in the image in figure 2C, where the lattice fringes of some single particles are visible and make it possible to determine their size and shape, The JEOL analyses found that the NP are all crystalline. The aggregation dimension and the size and shape of both types of nanoparticles were quite similar, most of them being around some tens of nm (20-40) with an ovoidal/round shape. The same shape and dimension are reported in the literature for the CuO NPs obtained from Sargassum longifolium using a similar green synthesis procedure [46].

In samples from RA (Fig 3A), the individual crystals appear to be more agglomerated, as if they had partially 'sintered' (word used here simply to indicate a greater interaction between the individual particles). The diffraction patterns, reported as inset in the A panels of Fig 3 and 4, confirm that only CuO is present, in agreement with the results of XRD.

The EDX spectra from the two types of nanoparticles are shown in panels B of Figure 3 and 4. The NP powders were dispersed on Ni grids for TEM observations, instead of common Cu grids, to avoid confusing the response of the sample with that of the support. EDX measurements confirm the presence of only Cu and O in the samples; spurious signals in the spectrum represent the background of the analysis chamber.

  1. Lines 185-186: Clarify whether the deposition of the sample for characterization was done at room temperature or higher temperatures.

The deposition for characterization was done at room temperature. We have inserted the clarification in the text in the section Materials and Methods at Lines 311-316:

“Talos high resolution TEM (Talos F200S G2, SEM FEG, Thermo Fisher Scientific, Waltham, MA, USA) equipped with an EDX detector has been utilized. Nanoparticles were sonicated in Eppendorf tubes containing ethanol for 20 min, they were then spotted on Au TEM grids and allowed to evaporate for 15 minutes at room temperature. We observed the morphology of single particles (Figure 2) and of aggregates and took the EDX spectra at 80 KeV (Figure 2).”